# Comparative Analysis of Lymphocyte Populations in Post-COVID-19 Condition and COVID-19 Convalescent Individuals

**DOI:** 10.3390/diagnostics14121286

**Published:** 2024-06-18

**Authors:** Luisa Berger, Johannes Wolf, Sven Kalbitz, Nils Kellner, Christoph Lübbert, Stephan Borte

**Affiliations:** 1Department of Infectious Diseases and Tropical Medicine, Hospital St. Georg, 04129 Leipzig, Germany; 2Department of Laboratory Medicine, Hospital St. Georg, 04129 Leipzig, Germany; 3ImmunoDeficiencyCenter Leipzig, Jeffrey Modell Diagnostic and Research Center for Primary Immunodeficiency Diseases, Hospital St. Georg, 04139 Leipzig, Germany; 4Division of Infectious Diseases and Tropical Medicine, Department of Medicine I, Leipzig University Medical Center, 04103 Leipzig, Germany

**Keywords:** SARS-CoV-2, COVID-19, immune system, lymphocyte subpopulations, long-COVID, post-acute COVID syndrome, post-COVID-19 condition (PCC)

## Abstract

Reduced lymphocyte counts in peripheral blood are one of the most common observations in acute phases of viral infections. Although many studies have already examined the impact of immune (dys)regulation during SARS-CoV-2 infection, there are still uncertainties about the long-term consequences for lymphocyte homeostasis. Furthermore, as persistent cellular aberrations have been described following other viral infections, patients with “Post-COVID-19 Condition” (PCC) may present similarly. In order to investigate cellular changes in the adaptive immune system, we performed a retrospective analysis of flow cytometric data from lymphocyte subpopulations in 106 patients with confirmed SARS-CoV-2 infection who received medical care at our institution. The patients were divided into three groups according to the follow-up date; laboratory analyses of COVID-19 patients were compared with 28 unexposed healthy controls. Regarding B lymphocyte subsets, levels of IgA + CD27+, IgG + CD27+, IgM + CD27− and switched B cells were significantly reduced at the last follow-up compared to unexposed healthy controls (UHC). Of the 106 COVID-19 patients, 56 were clinically classified as featuring PCC. Significant differences between PCC and COVID-19 convalescents compared to UHC were observed in T helper cells and class-switched B cells. However, we did not detect specific or long-lasting immune cellular changes in PCC compared to the non-post-COVID-19 condition.

## 1. Introduction

For more than four years, a global struggle was caused by the effects of the pandemic due to infections with severe acute respiratory syndrome coronavirus type 2 (SARS-CoV-2). Following the initial outbreak in 2019 in Wuhan, China, the virus rapidly spread into a pandemic with about 770 million confirmed cases and resulting in over 6.9 million deaths worldwide (until April 2024) [1].

Numerous studies have already shown that patients with a clinically severe course of COVID-19 present with abnormalities in several laboratory parameters, including lactate dehydrogenase (LDH), D-dimer, C-reactive protein (CRP), neutrophil counts, and pro-inflammatory cytokines, such as interleukin-6 and others [2]. Another common feature in patients with severe COVID-19 is lymphopenia with dramatically reduced numbers of T-, B-, and natural killer (NK) cells [3]. Lower proportions of these cells have been linked to disease severity, as reported in 96% of severe COVID-19 patients [4]. Lymphocyte subset counts may, therefore, provide prognostic information for COVID-19 disease severity and convalescence when considered in conjunction with other clinical information [5,6]. Furthermore, simultaneous recruitment of neutrophil granulocytes by number and function (NETosis) has been proposed to be relevant in COVID-19 outcomes.

Currently, the pathogenesis of reduced lymphocytes in COVID-19 patients is only partially understood. Some of the most common potential mechanisms reported to lead to lymphocyte deficiency are as follows:(i)Increased cellular death caused by the direct infection of SARS-CoV-2 Ribonucleic acid (RNA) in immune cells [7];(ii)Upregulation of the p53-mediated apoptosis signaling pathway in peripheral blood mononuclear cells (PBMC) [8];(iii)T cell extravasation and migration to inflamed tissue sites [4];(iv)Cytokine-storm-induced cellular apoptosis [9]; and(v)T lymphocyte exhaustion upon repeated activation [9].

Moreover, a relevant fraction of patients is prone to prolonged multi-organ complaints after the initial time of acute infection and illness. Under these long-term health consequences originating from COVID-19, the World Health Organization (WHO) proposed a clinical definition and the name “Post-COVID-19 Condition” (PCC) to unify various existing terms [1].

Of hospitalized patients, 87% have at least one persistent symptom at a mean of 60 days after symptom onset [10]. PCC has also been mentioned in a large number of studies, with a prevalence of up to 43% [11].

In this study, we aimed to evaluate changes in lymphocyte subpopulations with a particular focus on B cell subsets following SARS-CoV-2 infection in convalescent individuals and patients with PCC. Our evaluation may create new insights into the long-term consequences of SARS-CoV-2 infection and may provide a benefit for patients with persistent symptoms due to the lack of therapeutic alternatives.

## 2. Materials and Methods

### 2.1. Study Population

Between March 2020 and February 2021, 176 patients with PCR-confirmed SARS-CoV-2 infection and reappearance in the outpatient department (OPD) of our institution were consecutively enrolled. Patients with inborn errors of immunity or secondary immunodeficiencies, known autoimmune disorders or cancer, deceased patients, clinic staff members, and subjects with lacking data of flow cytometric analyses were excluded from the analysis afterwards.

In consideration of our selection criteria, 106 COVID-19 patients (aged between 32 and 88 years, 47% females) were included in this retrospective study. Demographics and clinical characteristics are shown in Table 1.

During medical aftercare, COVID-19 patients were asked about COVID-19-associated symptoms, already-known chronic diseases, and persistent symptoms, such as fatigue, cough, shortness of breath, and difficulties while thinking or concentrating. Considering the World Health Organization (WHO) guidelines, we distinguished between PCC and Non-Post-COVID-19 Condition (NPCC) patients [1]

In addition to a detailed case history including pre-existing medical conditions and COVID-19 severity, which were determined per criteria defined by the WHO [12], we checked the data of clinical examination and vital parameters (blood pressure, heart rate, oxygen saturation), which were collected as a part of routine medical aftercare.

For all patients routine laboratory analyses including blood type testing, complete blood counts including neutrophil-to-lymphocyte ratio (NLR), coagulation status (Quick/International normalized ratio (INR), thrombin time, fibrinogen, anti-thrombin III, fibrin monomers, D-dimer), serum protein electrophoresis, parameters to assess organ function, namely liver (bilirubin, alanine aminotransferase) and kidney (creatinine, glomerular filtration rate, urea), and inflammatory state (CRP, immunoglobulins) were performed. Furthermore, flow cytometric analyses of lymphocyte subpopulations were routinely performed as described below. No additional blood withdrawal for study purposes was required.

The COVID-19 patients were stratified into three groups according to the follow-up date: group 1 (G1): 85 to 150, group 2 (G2): 151 to 210, and group 3 (G3): 211 to 320 days after symptoms onset (Table 1).

A total of 28 healthy control subjects unexposed to SARS-CoV-2 (aged between 32 and 77 years, 75% females) visiting the Hospital St. Georg in Leipzig, Germany, between September 2017 and July 2018, with the same analyses of lymphocyte subsets by flow cytometry, were used as unexposed healthy controls (UHC) (Table 1). The UHC were selected out of 608 subjects due to the following exclusion criteria: patients with acute respiratory infections, inborn errors of immunity or secondary immune deficiencies, autoimmune disorders, those with transient hypogammaglobulinemia, leukocytosis/leukopenia and/or lymphocytosis/lymphopenia according to age-dependent cut-off values, and elevated CRP > 5 mg/L. In the case of follow-up/repeat measurements in the same subject, only the initial data were considered.

### 2.2. Whole-Blood Leukocyte Isolation and Cell Staining Procedures

From all individuals included in this study, 2.7 mL of peripheral venous blood was routinely sampled and analyzed. Before the leukocyte solutions were prepared, complete blood counts from a well-mixed and clotting-free Ethylenediaminetetraacetic acid (EDTA) blood sampling container were measured on an IVD-certified particle counter (Sysmex XN-2000, Norderstedt, Germany). The percentage values of the studied subsets obtained by cytometric reading were converted into absolute values (cells/µL) according to the absolute number of leucocytes (WBC) count provided by the hematological analyzer Sysmex XN-2000.

Afterward, red blood cells were lysed by applying an ammonium chloride-based reagent (BD Pharm Lyse™, BectonDickinson, Franklin Lakes, NJ, USA). This lysing solution was adjusted to 35 °C prior to use. After incubation for 15 min, centrifugation at 600× *g* for 10 min was performed. Subsequently, pelleted leukocytes were washed twice in phosphate-buffered saline (PBS), supplemented with 5% bovine calf sera (BCS), and resuspended in PBS/BCS. A volume of 100 µL of this leukocyte suspension was stained with two premixed antibody mixtures (Appendix A, Figure 1) for the differentiation and quantification of (i) general lymphocyte subsets (LS, seven color antibody mixture) and (ii) B cell subpopulations (BS, eight color antibody mixtures) for 15 min at room temperature in the dark. Representative scatter blots of the two flow cytometry panels and the applied gating strategy are shown in Figure 1. In addition to the usual B cell subpopulation studies, our BS panel considered the evaluation of CD21low B cells and transitional B cells. The latter are precursors of mature B cells, which are known to produce IL-10 and regulate CD4+ T cell proliferation and differentiation toward T helper (Th) effector cells [13]. CD21low B cells represent a unique population of memory B cells which increase during both aging and autoimmunity [14].

Another washing step with PBS/BCS was performed to remove unbound antibodies, and the pelleted cells were re-suspended in 300 µL of PBS/BCS. Prepared cell solutions were acquired on a BD FACSLyric™ flow cytometer (BD, Franklin Lakes, NJ, USA) within 30 min of staining using the acquisition settings detailed below:(i)LS: 3 min or 200,000 leucocytes;(ii)BS: 3 min or 20,000 CD19 + CD20+ B cells.

### 2.3. Flow Cytometry Analysis Software

A BectonDickinson (BD) FACSLyric™ instrument with three lasers (red, blue, and violet) was used for flow cytometric measurements. BD FACSuite™ 1.5 software was utilized for assay generation and data acquisition. Automated cytometer setup, assay setup, and performance tracking were performed with the daily acquisition of BD CS&T™ IVD beads according to the supplier’s instructions.

### 2.4. Statistical Analysis

Statistical analyses and visualization of the data were performed in Jupyter Notebook using Python 3.9.12. The packages pandas v.1.4.2., seaborn v.0.11.2, and matplotlib v.3.5.1 were used for data handling and the creation of scattered boxplots, and statistical analyses were performed using scipy v.1.7.3. Data were tested for a Gaussian distribution using the Shapiro–Wilk normality test. When a Gaussian distribution was not confirmed, the groups were tested for significant differences using the Mann–Whitney U test. Fisher’s exact test was applied to compare categorical variables. We refer to statistical significance the following way: *p* < 0.05 represented with one star, *p* < 0.01 represented with two stars, and *p* < 0.001 represented with three stars.

## 3. Results

### 3.1. Demographic and Clinical Characterization of the Study Population

A total of 106 patients entered the study based on the inclusion and exclusion criteria. They were compared to healthy donors who were included before December 2019 and thus were considered unexposed to SARS-CoV-2 (UHC). The mean age of the COVID-19 cohort was 60.7 years (G1: 66.4 years, G2: 65.2 years, G3: 52.3 years), whereas the mean age of the UHC cohort was 49.6 years. Further baseline demographic data and gender distribution are summarized in Table 1.

In total, 44 (40.9% females) of 106 patients (41.5%) were asymptomatic or had mild symptoms during the acute phase of SARS-CoV-2 infection (according to WHO clinical progression scale 2–3). Otherwise, 62 (50% female) patients (58.5%) were hospitalized with moderate or severe symptoms (according to the WHO scale 4–5). In contrast to the acute phase, which was partially characterized by lymphopenia, an elevation of the neutrophil-to-lymphocyte ratio (NLR) and C-reactive protein (CRP, see Appendix A) as well as increased LDH and pro-inflammatory cytokines, we detected a normalization of lymphocytes, NLR, and biochemical and inflammatory markers in each follow-up.

### 3.2. Lymphocyte Phenotype of the Study Cohort

A persistent decrease in Natural Killer (NK) cells in G1 (*p* < 0.003) and G2 of COVID-19 convalescents (*p* < 0.002) was observed. NKT were significantly reduced in G2 compared to UHC (*p* < 0.029) (Figure 2).

Considering B cell subsets, we observed decreased levels of Immunoglobulin A (IgA+) CD27+ (*p* < 0.034), IgG + CD27+ (*p* < 0.006), and IgM + CD27− B cells (*p* < 0.034) in G3 in comparison to UHC. In contrast, a decrease in switched B cells was found in G1 (*p* < 0.048) and G3 in relation to UHC (*p* < 0.01). Transitional B cells were elevated in G1 (*p* < 0.046) and G2 (*p* < 0.045) compared to UHC in opposition to CD21low b cells, which showed no significant shift (Figure 2). Furthermore, we noticed relevant numbers of outliers for both CD138+ plasma cells and CD38++ plasma blasts. In contrast to plasma cells, plasma blasts significantly increased in each follow-up, whereas the first revealed no significant change.

### 3.3. Occurrence and Cellular Impact of PCC in the Study Group

Of 106 COVID-19 patients (52% female), 56 were designated as PCC based on the occurrence of one of three major symptoms (fatigue, dyspnea, or chest pain) [15]. In detail, 57.1% of patients in G1 (50% female), 65.2% in G2 (50% female), and 35.9% in G3 (57% female) affirmed persistent symptoms. The median age was 64 years in the PCC cohort and 56 years in the NPCC cohort. Regarding laboratory parameters (e.g., leukocytes, platelets, hemoglobin, and C-reactive protein), a normalization in the median values established the age-specific threshold value (PCC/NPCC, Appendix A). Because of the initial decrease in lymphocyte numbers during the acute phase, which was still detectable in G1 (median PCC 1.67 Gpt/L; median NPCC 2.08 Gpt/L) and not detectable in G2 (median PCC 1.77 Gpt/L; median NPCC 1.73 Gpt/L) and 3 (median PCC 1.85 Gpt/L; median NPCC 1.83 Gpt/L), we further analyzed lymphocyte subset cells in peripheral blood (Appendix A). There were no significant differences between the PCC and NPCC cohorts in the T cell lymphocyte as well as B cell subsets, NKT, and NK cells (Appendix A).

In comparison to UHC, we observed decreased levels of B cells (PCC median 150/µL, *p* < 0.005; NPCC median 250/µL), IgA + CD27+ B cells (PCC median 9/µL, *p* < 0.048; NPCC median 8/µL), and IgG + CD27+ B cells (PCC median 5/µL, *p* < 0.003; NPCC median 6/µL) in PCC relative to NPCC in G3. This variation could not be confirmed in the earlier follow-ups (G1 and G2). Non-significant differences between PCC and NPCC relative to UHC were observed in switched memory B cells in our PCC cohort at all follow-up times (G1: PCC median 40/µL, *p* < 0.089, G2: PCC median 45/µL, *p* < 0.164). In G3, the difference was significant (PCC median 29/µL, *p* < 0.004). Further results are summarized in Figure 2 and Figure 3.

## 4. Discussion

Recovery from SARS-CoV-2 infection is often associated with persistent symptoms months after infection, including fatigue, weakness, and shortness of breath [10,16]. Several recent studies suggest ongoing immune dysregulation in COVID-19 convalescents while profiling the immune system using multi-parameter flow cytometry, bulk and single-cell transcriptomics with a focus on T cells [17,18]. Using flow cytometry and immunological and serological assays, we provide a comprehensive look at additional immune cell subsets, especially B cell subsets, after COVID-19. To capture the complete range of immune recovery, we included the full spectrum of disease, with severity ranging from asymptomatic infections to mild disease managed in the community to requirements for ICU care. Moreover, we observed changes in symptom characteristics and clinical parameters over time and drew our results in comparison to UHC to point out potential abnormalities.

First, compared to previously published data, we observed a normalization of clinical parameters, e.g., acute-inflammatory biomarkers (CRP, LDH, neutrophils, lymphocytes), which are known to be elevated or decreased in the acute phase [2,19]. The NLR, which is an established biomarker for the prediction of progression or severity for COVID-19 [20], exceeded the proposed threshold for disease severity of ≥4.5 in 44.7% of the COVID-19 cases in the acute phase. Otherwise, the NLR normalized in the follow-up independently to time or present post-COVID syndrome. Another known alteration during acute COVID-19 is the depletion of T cells and their subsets, as well as NK and NKT cells [17,21]. With regard to our data, the examined innate immune cell subsets returned to baseline in convalescents within about three months, except for NK cells. Herein, normalization of these cells has only been observed in the latest follow-up. This elevation of NK cells led to the assumption that there could be increased cell proliferation after acute COVID-19. Otherwise, viral infections are often associated with high levels of NK cells, which may indicate a persistent elevation. In general, the main impact of NK and NKT cells in COVID-19 pneumonia has been discussed [22]; hence, these alterations are indicative of prolonged dysregulation in immune cells.

The number and function of neutrophil granulocytes have furthermore been implicated in the course and outcome of COVID-19 infection. In our current investigation, which mainly focused on lymphocytes, we were, however, not able to delineate a specific predictive value of the neutrophil-to-lymphocyte ratio on either the course or the outcome of the studied cohorts.

Additionally, altered frequencies of adaptive immune cell populations, including activation of cytotoxic T (CD8+) and T helper (CD4+) cells in recovered patients (G3), were significantly changed compared to UHC. These results agree with another study that showed increased cell proliferation of CD4+ and CD8+ T cells after SARS-CoV-2 infection [23]. In contrast, we noticed a normalization of these cells in G1 and G2, which could be explained by the mild disease severity of our study population, as well as the late start of our observations (85 days after symptom onset).

Another notable driving factor in the heterogeneous immunological responses observed in COVID-19 could be immunosenescence and inflammaging. Immunosenescence refers to the aging of the immune system, which is mainly characterized by a decrease in naïve T cell counts together with an accumulation of CD4+ and CD8+ memory and terminal effector T cells. This may lead to increased vulnerability to infections and an impaired response to vaccination [24]. Recent publications have presumed that SARS-CoV-2 infection could lead to accelerated immunosenescence in distinct populations [25].

With regard to age-related alterations in B cells and the significant distribution of the mean age in our cohort and UHC, we have to point out a relatively newly defined subpopulation: CD21low B cells (sometimes referred to as age-associated B cells). CD21low B cells are a naturally occurring population of antigen-experienced B cells that expand continuously with age in healthy individuals but accumulate prematurely in patients with autoimmunity, inborn errors of immunity, and/or infectious diseases [14]. Wildner et al. also observed an expansion of CD21–/lowCD27– cells in critically ill COVID-19 patients, wherein the number and proportion decreased in patients who recovered [26]. Although there was no significant difference between our study population and UHC in CD21low B cells, there is a considerable trend in G1 and G2, the two groups with the highest mean age.

Moreover, there is ample evidence that CD21low B cells contribute to the production of both disease-specific antibodies and pathogenic autoantibodies. In several conditions, they, or subsets thereof, are associated with key disease manifestations, e.g., in systemic lupus erythematosus where the frequency of these cells correlates with autoantibody levels and disease activity score as well as rheumatoid arthritis with joint destruction [27]. Here, we made no distinction between the subsets of CD21low B cells. In order to better understand the role and nature of these cells after the acute phase of COVID-19, further investigations are needed.

In addition, Ryan et al. reported that B cell activation/exhaustion markers remain elevated following SARS-CoV-2 infection [28]. Our observations showed entirely recovering B cell counts, which is consistent with other data [17,22]. Regarding the antiviral immune response, we focused on switched memory B cells that result from the maturation of B cells. Compared to UHC, we observed significant exhaustion of switched B cells in two groups (G1 and G3) (Figure 3). Contrasting publications have marked a significant increase in memory B cells after SARS-CoV-2 infection, which exhibited protective antiviral functions [28,29,30]. Remarkable other studies have been pointing out that antigen-specific responses to SARS-CoV-2 can persist for several months after infection [17,22]. The reasons for these unexpected results might be diverse. SARS-CoV-2 can induce host cell death via different pathways, i.e., apoptosis in response to viral infection or over-activation, necroptosis, pyroptosis, and PANoptosis [31].

As outlined in another study, different levels of COVID-19 severity could be associated with different levels of immune memory and subsequent immune protection [32]. In addition, the dysfunction of switched memory B cells after infection can be discussed. Generally, it is well recognized that heterogeneity is a central future of immune memory in SARS-CoV-2 [29].

Further classification of B cell subsets into transitional (CD38 + IgM++) and IgA+/IgG+ CD27+ and IgM + CD27− B cells revealed a significant expansion of transitional B cells in G1 and G2. This could be caused by the significant reduction in transitional B cells in acute COVID-19, which is characterized by a temporary increase in convalescence. At present, related observations revealed an impairment of immune-regulatory functions of transitional B cells in various immune diseases like autoimmune rheumatic diseases and neuro-immunological diseases [33]. An expansion of transitional B cells is also frequently reported in patients with systemic lupus erythematosus (SLE) and Sjögren’s syndrome (SS) which strengthens the hypothesis that autoimmunity has an impact on PCC.

In addition, current studies have observed that virus-specific antibodies are detectable for several months after recovery [22,29,32]. As expected, our findings demonstrated detectable IgG and IgA antibody responses against SARS-CoV-2 after infection. Although we found no remarkable difference in the amount of IgA + CD27+ B cells in convalescents and UHC in G1 and G2, the reduction in G3 was significant and in accord with other findings [18]. In addition, sustained production of neutralizing IgG+ virus-specific antibodies has been consistently correlated with protection from virus infection [34]. The drop of IgG + CD27+ B cells in our latest follow-up could lead to the assumption that the concentration of IgG+ antibodies decreases within 7–11 months after infection.

In agreement with a previously published study, the B cell profiles of convalescent plasma donors after COVID-19 disease frequently differs [30]. Examining B cell phenotypes in convalescent patients highlights that alterations in B cell subsets during severe acute COVID-19 are largely restored upon convalescence. These viral-neutralizing antibodies are secreted by plasma cells to provide durable protection after infection. Further studies revealed that circulating antibody-secreting cells (ASC) defined as CD19 + CD27hiCD38hiCD138+ were expanded in severe SARS-CoV-2 infections, although their occurrence was not associated with virus-specific IgM [35]. In convalescents, we detected a normalization of these cell subsets. Hence, the rapid antibody decay is a manifestation of apoptosis of the nascent blood ASC.

Individuals suffering from PCC demonstrated subtle differences in immune responses compared with those without persistent symptoms. Risk factors for PCC may include age, more severe acute infection, socioeconomic factors, and female sex [36]. Given the high prevalence (51%) of persistent symptoms among our study population, we evaluated immune responses in individuals with or without PCC and identified no significant differences but rather trends that should be mentioned. Indeed, B cells are known to be reduced in the acute phase of SARS-CoV-2 infection [3,37]. Moreover, Hu F. et al. have reported a correlation between decreased numbers of B and T cells and persistent SARS-CoV-2 shedding, which may further perpetuate chronic immune activation in PCC [38]. Although there was no significant difference between NPCC and PCC, we found reduced amounts of B cells in PCC patients in comparison to NPCC patients (G1: *p <* 0.126, G2: *p* < 0.596, G3: *p* < 0.087). Whether reduced numbers of B cells could influence the pathophysiology of PCC has not yet been proven, although their impact on COVID-19 disease has been discussed [39]. Another trend is noticeable in switched memory B cells (CD21 + CD27+); here, we observed a reduction of these cells in PCC compared to NPCC, especially in G3 (with no significant value). Additionally, switched memory B cells were significantly decreased in PCC compared to UHC. Nevertheless, the variability in the penetrance of this phenotype has already been mentioned by others and may be explained by the influence of gender and pre-existing autoimmunity [30].

Interestingly, many of our PCC patients met the diagnostic criteria for myalgic encephalomyelitis/chronic fatigue syndrome (ME/CFS), a neuroinflammation-linked condition characterized by a range of debilitating chronic symptoms [40,41]. Overlap between PCC and ME/CFS diagnoses is not surprising since numerous cases of ME/CFS begin with a viral infection, e.g., Epstein–Barr Virus (EBV) [42,43] and Human Herpesvirus-6 (HHV6) [44]. Consistent abnormal findings in ME/CFS include T cell exhaustion and other T cell abnormalities, diminished NK cell function, mitochondrial dysfunction, and vascular and endothelial abnormalities [45], which have been observed in PCC patients as well [46].

Furthermore, the presence and reactivation of chronic viral infections, such as cytomegalovirus (CMV), EBV, and human immunodeficiency virus (HIV), have been proposed as potential contributors to PCC [47]. Serological evidence in a prior study has already suggested that recent EBV reactivation is associated with a higher likelihood of developing PCC [47]. Notably, in this study, we did not screen for coincidental viral infections after SARS-CoV-2 infection.

Historically suggested, viral infections have had a complex relationship with a variety of autoimmune diseases, e.g., rheumatoid arthritis (RA), systemic sclerosis (SS), and systemic lupus erythematosus (SLE). Several autoimmunity phenomena were observed during acute COVID-19, e.g., developing new IgG- autoantibodies in hospitalized patients [48] producing autoantibodies against immunomodulatory proteins, including cytokines, chemokines, and cell surface proteins [49]. These features raise questions about whether they result from an immune inflammatory or autoimmune process triggered by COVID-19, and whether they are truly novel syndromes or are characteristics of previously described post-viral inflammatory syndromes. This autoimmune hypothesis could justify the higher incidence of this syndrome in women, which mostly conforms to our findings.

Currently, the SARS-CoV-2 pandemic seems to be under control in most countries, and the consequences of SARS-CoV-2 infection have come to the fore. Indeed, research into PCC has accelerated, but existing research is not enough to improve the outcomes for people who are suffering from PCC.

Compared to other studies, the sample size of our study is limited, particularly in the case of patients with more severe diseases. This is important given the apparently highly heterogeneous recovery in immune dysregulation over time. In addition, the mean age of our control cohort was much lower than in our study population, which could have an impact on lymphocyte counts. Another limiting factor might be the lack of inclusion of patients with non-European ancestry. It is also noteworthy to consider the lack of differentiation in circulating virus variants, which could have an influence on the pathogenesis. While our flow cytometry analyses enabled the assessment of multiple parameters, they did not include markers for neutrophils, monocytes, lymphokines, or dendritic cells (DC), which are altered in COVID-19 convalescents, as shown in a previous study [49]. Further larger studies are needed to more fully assess the differences due to disease severity, comorbidities, treatment, and other confounders.

## 5. Conclusions

In this retrospective study, we aimed to determine potential changes in the immune cells of convalescent COVID-19 patients with a focus on B cells. Many alterations, which are representative of acute COVID-19 (e.g., lymphopenia), returned to baseline in convalescence. About half of our study population complained of persistent symptoms after SARS-CoV-2 infection. Although there might be an association between disease severity, age, and immune cell abnormalities in the acute phase, we could not identify an association between PCC, age, and gender, as shown in other studies [30].

In addition, no definite immune dysregulation/abnormality was found in patients suffering from persistent symptoms compared to NPCC, although some B cell alterations in PCC are characteristic of autoimmune diseases. This suggests that more in-depth immunophenotyping combined with transcriptome investigations could provide further clues to the biological mechanisms underlying this poorly understood condition, which is of growing clinical concern.

## Figures and Tables

**Figure 1 diagnostics-14-01286-f001:**
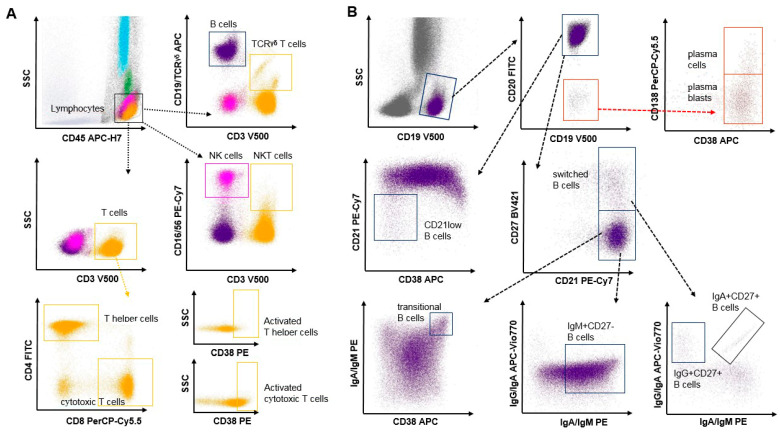
Representative gating for detection of T, B, and natural killer (NK) cells (**A**), as well as B cell subpopulations (**B**). For quantification of lymphocyte subsets, fluorochrome-conjugated monoclonal antibodies against CD45, CD3, CD4, CD8, CD16/CD56, T cell receptor (TCR) γδ, CD19, and CD38 were used. For gating to determine proportions of B cell subpopulations, monoclonal CD19, CD20, CD38, CD138, CD21, CD27, IgG, IgA, and IgM antibodies were applied.

**Figure 2 diagnostics-14-01286-f002:**
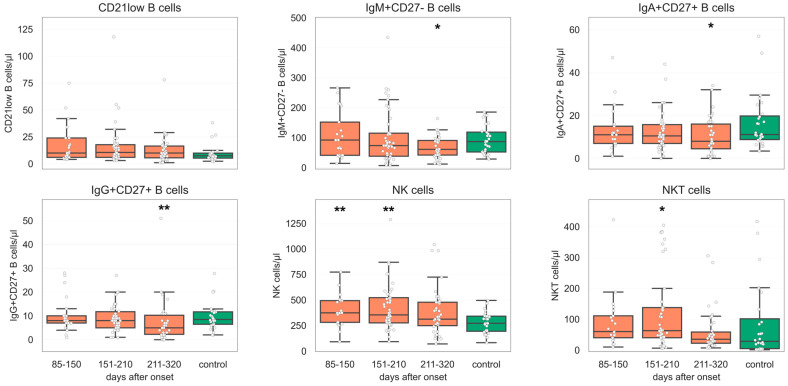
Box plots of the 25th to 75th percentile of B cell subpopulation counts (CD21 + low B, IgG + CD27+ B, IgA + CD27+ B, and IgM + CD27+ B cells) as well as Natural Killer (NK) cells and natural killer T cells (NKT). The middle line represents the median, and the upper/lower whiskers represent the max/min value within 1.5× the 75th/25th interquartile range, respectively. N = 28 healthy individuals, group 1 (G1) = 85–150 days after symptom onset (n = 21), group 2 (G2) = 151–210 days after symptom onset (n = 46), group 3 (G3) = 211–320 days after onset (n = 39). Statistical testing was performed using the Shapiro–Wilk normality test and Mann–Whitney U test. * *p* ≤ 0.05, ** *p* ≤ 0.01.

**Figure 3 diagnostics-14-01286-f003:**
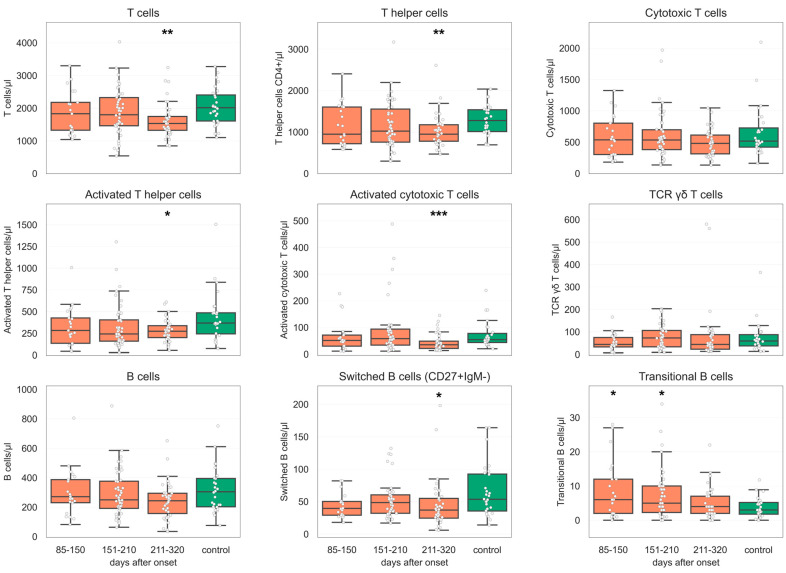
Box plots of the 25th to 75th percentile of T and B cell counts, as well as T cell subpopulations (T helper, cytotoxic T, activated T helper, activated cytotoxic T, and T cell receptor [TCR] γδ T cells), switched B cells, and transitional B cells. The middle line represents the median, and the upper/lower whiskers represent the max/min value within 1.5× the 75th/25th interquartile range, respectively. N = 28 healthy individuals, group 1 (G1) = 85–150 days after symptom onset (n = 21), group 2 (G2) = 151–210 days after symptom onset (n = 46), group 3 (G3) = 211–320 days after onset (n = 39). Groups 1-3 are colored orange, control group is depicted in green. Statistical testing was performed using the Shapiro–Wilk normality test and Mann–Whitney U test. * *p* ≤ 0.05, ** *p* ≤ 0.01, *** *p* ≤ 0.001.

**Table 1 diagnostics-14-01286-t001:** Demographic characteristics and laboratory results of controls and COVID-19 cases (divided into three follow-up groups).

	Controls (UHC)	COVID-19 All	COVID-19 Group 1 (G1)	COVID-19 Group 2 (G2)	COVID-19 Group 3 (G3)
N	28	106	21	46	39
Females (%)	75	47.1	47.6	50.0	41.0
Mean age (min–max)	48.9 (32–72)	60.6 (32–88)	66.4 (37–84)	65.2 (32–88)	52.3 (32–81)
Blood sampling after symptoms onset (days)	n/a	85–319	85–150	151–210	211–320
Severity of disease (1) Mild (WHO 2–3) (2) Moderate/severe (WHO 4–7)	n/a n/a	44 (41.5%) 62 (58.5%)	4 (19.1%) 17 (81.0%)	15 (32.6%) 31 (67.4%)	25 (64.1%) 14 (35.9%)
Positive SARS-CoV-2 serology	n/a	106	21	46	39
Post-COVID-19 condition	n/a	56 (51%)	12 (57%)	30 (65%)	14 (35.9%)
Median leucocytes in cells per µL (Min/Max)	7.41 (4.25/10.71)	6.59 (3.68/12.93)	7.41 (4.06/11.4)	6.50 (3.68/11.7)	6.10 (4.29/12.93)
Median lymphocytes in cells per µL (Min/Max)	2.08 (1.31/4.27)	1.84 (0.9/3.57)	1.91 (1.23/3.56)	1.76 (0.92/3.22)	1.84 (0.9/3.57)
Median neutrophils in cells per µL (Min/Max)	4.6 (2.43/8.21)	4.05 (1.44/10.7)	4.68 (1.84/7.06)	3.87 (1.79/9.65)	3.84 (1.44/10.7)
Neutrophil lymphocyte Ratio (Min/Max)	2.04 (0.78/4.12)	2.07 (0.61/7.34)	1.89 (0.97/4.73)	2.14 (1.03/7.34)	2.04 (0.62/7.27)
Median hemoglobin in mmol/L (Min/Max)	n/a	8.9 (5.2/11.5)	9.1 (6.8/10.9	8.7 (5.2/10.7)	9.05 (7.6/11.5)
Median platelets in Gpt/L (Min/Max)	n/a	236 86/435)	204 (123/359	242 (86/435)	237 (154/402)
C-reactive Protein in mg/L (Min/Max)	1.7 (0.3/9.3)	1.6 (0.3/21.3)	1.6 (0.3/14.0)	1.6 (0.3/21.3)	1.5 (0.3/10.0)
Lactate dehydrogenase in µmol/L*s (Min/Max)	n/a	3.36 (2.39/6.3)	3.45 (2.78/4.76)	3.36 (2.39/5.34)	3.29 (2.52/6.3)
Median IgG in g/dL (Min/Max)	10.45 (7.6/18.2)	10.8 (1.8/28.9)	9.25 (6.6/13.5)	11.25 (1.8/28.9)	10.3 (6.8/17.0)
Median IgA in g/dL (Min/Max)	1.965 (0.85/4.15)	2.38 (0.05/7.04)	2.47 (0.81/5.64)	2.42 (0.05/7.04)	2.34 (0.7/5.68)
Median IgM in g/dL (Min/Max)	0.975 (0.31/12.5)	0.87 (0.25/2.75)	0.765 (0.27/1.79)	0.955 (0.34/2.75)	0.800 (0.25/2.64)

## Data Availability

The raw data supporting the conclusions of this article will be made available by the authors upon request.

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
