# Peer review of "Comparative Analysis of Lymphocyte Populations in Post-COVID-19 Condition and COVID-19 Convalescent Individuals"

_diagnostics, 2024, doi:10.3390/diagnostics14121286_

Round 1
Reviewer 1 Report
Comments and Suggestions for Authors
This paper collected patients' data to explore potential remarkable laboratory changes among patients suffered Post-Covid-19 conditions.
Overall, the paper was written very well. The authors provided good background information, described the lab method in detail, and had an inspirated discussion part. But there are some small issues here and there, I hope the authors could address before getting accepted for publication.
1. About the results demonstrated in Figure 1, is that from 1 subject, or grouped data? If it is from a single sample, why this is chosen? Can it represent the whole study population?
2. Please specify statistical significance level in Statistical Analysis section.
3. Table 1 need formatting update. Please remove all vertical lines and unnecessary horizonal lines.
4. How is Group 1, 2, and 3 were divided? Please describe ahead.
Author Response
- About the results demonstrated in Figure 1, is that from 1 subject, or grouped data? If it is from a single sample, why this is chosen? Can it represent the whole study population?
Thank you for the opportunity to specify our results. We inserted Figure 1 into the manuscript to better illustrate the gating strategy of our two flow cytometry panels which are described in the method section. The depicted dot blots represent the lymphocyte subsets including plasma blasts and cells (cells which are found rarely in blood samples of healthy subjects) of one COVID-19 patient. During the preparation of the manuscript we have discussed the usage of merged scattergrams to compare healthy controls and the three COVID-19 groups as you propose. Due to the minor differences between the groups we decided to use box blots intead. We have include the following sentence to underline the reason for inclusion of Figure 1:
Representative scatter blots of the two flow cytometry panels and the applied gating strategy are shown in Fig.1 (Page 4, paragraph 2, line 143-144)
- Please specify statistical significance level in Statistical Analysis section.
We totally agree with the reviewer that we missed to specify statistical significance levels. Therefore, we added them in page 5, paragraph 3, line 174-176.
- Table 1 need formatting update. Please remove all vertical lines and unnecessary horizonal lines.
Thank you for pointing this out. We optimized the format of „table 1“ (page 6).
- How is Group 1, 2, and 3 were divided? Please describe ahead.
The reviewer made a valid point that the paper should include the information how we divide our population into three groups. We revised the paragraph (page 3, paragraph 6, line 113-114).
Reviewer 2 Report
Comments and Suggestions for Authors
This paper by Berger et al., entitled “Comparative analysis of lymphocyte populations in Post- COVID-19 Condition and COVID-19 convalescent individuals” tried to investigate cellular changes in the adaptive immune system by a retrospective analysis of flow cytometric data from lymphocyte subpopulations in 106 patients with confirmed SARS-CoV-2 infection, receiving medical care. Clinical and laboratory data during the acute phase of Covid-19 infection should be provided to both better characterize potential heterogeneity of the study population and show the clinical profile of patients over time.
Specific Comments
1. Covid-19 infection has been reported to be accompanied by a derangement of the cross-talk between innate immunity (managed by neutrophils) and adaptive immunity (managed by lymphocytes). Refer for details to a review by Buonacera et al., Int J Mol Sci 2022. Do Authors have any data on neutrophils both in the acute phase and in the post-Covid one?
2. Inflammation and its markers, such as NLR and CRP, are known to underlie Covid-19 infection in the acute phase, so that their relationships with prognosis has been brought to the fore to predict ICU admission (Regolo et al. J Clin Med 2022) and survival related to respiratory failure (Regolo et al. J Clin Med 2023). Although patients with respiratory failure were excluded, what about NLR and CRP in your cohort during the acute phase, as compared with the long term follow-up? Was there any relationnship with the behaviour of lymphocyte subpopulations? By the way, data on neutrophils are missing at all in Table 1.
3. The method of patients selection should be given. Were they consecutively included in this study, although retrospectively?
4. Although lymphopenia is a common feature of viral infections, by restricting the focus only on lymphocytes it could be taken the risk of obtaining inclonclusive results, particularly in Covid-19 patients, where the simultaneous involvement of neutrophil activation plays a major role in inducing lymphopenia also through extracellular trapping of neutrophil DNA fragments triggered by neutrophil activation (Kourilovitch & Maldonado, J Trans Autoimmun 2023).This issue should be highlighted throughout the manuscript to avoid that readers of Diagnostics, that is a journal largely open to clinicians, could erroneously catch the message that the study of lymphocyte subpopulations could help to characterize the follow-up of Covid-19 patients. In other words, the pattern of lymphocyte subpopulations, taken as alone, would not help to provide the readers with additional information on the complex immunologic dysfunction induced by Covid-19 infection.
5. Limitations of the study are missing in the final part of the Discussion.
6. Overall, clinical and laboratory data from the beginning throughout all the course of Covid-19 patients history, including outcome, is needed. In other words, it is difficult to draw conclusions regarding the post-Covid phase if we do not know some information regarding both the acute phase and the final outcome, given that in Table 1 a different disease’s severity was shown.
Author Response
- Covid-19 infection has been reported to be accompanied by a derangement of the cross-talk between innate immunity (managed by neutrophils) and adaptive immunity (managed by lymphocytes). Refer for details to a review by Buonacera et al., Int J Mol Sci 2022. Do Authors have any data on neutrophils both in the acute phase and in the post-Covid one?
The primary aim of our small study was to evaluate the distribution of lymphocyte subsets in COVID-19 with/without Post-COVID syndrome. Otherwise the impaired crosstalk between innate and adaptive play a significant role in the pathogenesis of COVID-19. Therfore, we have inserted neutrophil count and NLR (Median and Min/Max) into Table 1 for the considered disease and control groups as you suggested.
We also considered neutrophils and NLR at the acute phase (if available since approx. 1/3 of COVID-19 subjects were outpatients without blood sampling at the acute phase) in comparison to follow-up. As expected, we found significant differences in lymphocytes and NLR between acute phase and follow-up underline the restoring capacity of the lymphocytes. These data are now represented in supplementary table 2.
Further we have added the following to the discussion section (page 9, paragraph 2, line 260-266).
- Inflammation and its markers, such as NLR and CRP, are known to underlie COVID-19 infection in the acute phase, so that their relationships with prognosis has been brought to the fore to predict ICU admission (Regolo et al. J Clin Med 2022) and survival related to respiratory failure (Regolo et al. J Clin Med 2023). Although patients with respiratory failure were excluded, what about NLR and CRP in your cohort during the acute phase, as compared with the long term follow-up? Was there any relationnship with the behaviour of lymphocyte subpopulations? By the way, data on neutrophils are missing at all in Table 1.
Thank you for the reviewer´s constructive comment on inflammation and it´s markers in COVID-19. Regolo et al brilliantly highlighted the importance of biomarkers of inflammation e.g. PCT, CRP and NLR in the assessment of COVID-19-progression.
Indeed, several studies prooved elevated levels of LDH, CRP and NLR in acute COVID-19, in contrast we later detected a normalization of these cells. We specified the paragraph in our results (page 7, paragraph 1, line 192-196)
In contrast to the studies of Regolo et al. mentioned above we kept the focus on lymphocytes and their subpopulation. In consideration of the appreciable significance of neutrophils and NLR in acute COVID-19, we added them in „table 1“.
Generally this point is already adressed previously by including Supplementary Table 2 and a new part in the discussion section.
- The method of patients selection should be given. Were they consecutively included in this study, although retrospectively?
Thank you for the opportunity to optimize our method section by concretizing our patients selection. Indeed, at first all patients were consecutively included in the study, considering our retrospective selection criteria certain patient groups were excluded subsequently, we modified this point in our manuscript (page 3, paragraph 1-2, line 88-93).
- Although lymphopenia is a common feature of viral infections, by restricting the focus only on lymphocytes it could be taken the risk of obtaining inclonclusive results, particularly in Covid-19 patients, where the simultaneous involvement of neutrophil activation plays a major role in inducing lymphopenia also through extracellular trapping of neutrophil DNA fragments triggered by neutrophil activation (Kourilovitch & Maldonado, J Trans Autoimmun 2023).This issue should be highlighted throughout the manuscript to avoid that readers of Diagnostics, that is a journal largely open to clinicians, could erroneously catch the message that the study of lymphocyte subpopulations could help to characterize the follow-up of Covid-19 patients. In other words, the pattern of lymphocyte subpopulations, taken as alone, would not help to provide the readers with additional information on the complex immunologic dysfunction induced by Covid-19 infection.
We acknowledge the relevant comment indicate by the reviewer and have inserted highlighted notes regarding the relevance of neutrophil granulocyte number and function throughout introduction, methods and discussion.
- Limitations of the study are missing in the final part of the Discussion.
We sincerely apologise that the limitations of our study were missing as a separate paragraph in the discussion. We totally agree with you and revised the final part (page 13, paragraph 2, line 408-416).
- Overall, clinical and laboratory data from the beginning throughout all the course of Covid-19 patients history, including outcome, is needed. In other words, it is difficult to draw conclusions regarding the post-Covid phase if we do not know some information regarding both the acute phase and the final outcome, given that in Table 1 a different disease’s severity was shown.
We acknowledge the relevance of reviewer`s suggestion to provide more characterizing clinical and laboratory data with regard to outcome parameters of the patients included and cohorts analysed. We have re-iterated, that the study cohort presented herein was clinically characterized based on established WHO scoring system, as well as is representative of a typical regional patient distribution recruited in the acute phase of COVID-19 infection.
To further illustrate laboratory information, we have provided – as requested - an additional Supplementary Table 2 depicting the comparison of laboratory data assessed during the acute phase and post-COVID phase.
In a second separate Supplementary Table 3 we were able to compare Post-COVID patients and non-post-COVID cases to further draw conclusions on the final outcome of patients, which did not differ significantly based on the additionally requested laboratory information
Round 2
Reviewer 2 Report
Comments and Suggestions for Authors
All concerns were satisfactorily answered.
Comments on the Quality of English LanguageA revision of English should be recommended.